# DermaMamba: A Dual-Branch Vision Mamba Architecture with Linear Complexity for Efficient Skin Lesion Classification

**DOI:** 10.3390/bioengineering12101030

**Published:** 2025-09-26

**Authors:** Zhongyu Yao, Yuxuan Yan, Zhe Liu, Tianhang Chen, Ling Cho, Yat-Wah Leung, Tianchi Lu, Wenjin Niu, Zhenyu Qiu, Yuchen Wang, Xingcheng Zhu, Ka-Chun Wong

**Affiliations:** 1Department of Computer Science, City University of Hong Kong, Hong Kong SAR, China; zhongyyao2-c@my.cityu.edu.hk (Z.Y.); zliu39-c@my.cityu.edu.hk (Z.L.); thchen7-c@my.cityu.edu.hk (T.C.); ken.cho@my.cityu.edu.hk (L.C.); dlyatwah2-c@my.cityu.edu.hk (Y.-W.L.); tianchilu4-c@my.cityu.edu.hk (T.L.); wjniu2-c@my.cityu.edu.hk (W.N.); zhenyuqiu4-c@my.cityu.edu.hk (Z.Q.); wangyuchen.cs@my.cityu.edu.hk (Y.W.); 2Department of Mathematics, University College London, London WC1E 6BT, UK; yanyu.yan.24@ucl.ac.uk; 3Department of Clinical Laboratory, The Second People’s Hospital of Qujing City, Qujing 655099, China; 15987459671@163.com

**Keywords:** skin lesion classification, Vision Mamba, state space models, dual-branch fusion, medical image analysis, dermatology AI, melanoma detection, linear complexity, clinical decision support, ABCDE rule integration

## Abstract

Accurate skin lesion classification is crucial for the early detection of malignant lesions, including melanoma, as well as improved patient outcomes. While convolutional neural networks (CNNs) excel at capturing local morphological features, they struggle with global context modeling essential for comprehensive lesion assessment. Vision transformers address this limitation but suffer from quadratic computational complexity O(n^2^), hindering deployment in resource-constrained clinical environments. We propose DermaMamba, a novel dual-branch fusion architecture that integrates CNN-based local feature extraction with Vision Mamba (VMamba) for efficient global context modeling with linear complexity O(n). Our approach introduces a state space fusion mechanism with adaptive weighting that dynamically balances local and global features based on lesion characteristics. We incorporate medical domain knowledge through multi-directional scanning strategies and ABCDE (Asymmetry, Border irregularity, Color variation, Diameter, Evolution) rule feature integration. Extensive experiments on the ISIC dataset show that DermaMamba achieves 92.1% accuracy, 91.7% precision, 91.3% recall, and 91.5% mac-F1 score, which outperforms the best baseline by 2.0% accuracy with 2.3× inference speedup and 40% memory reduction. The improvements are statistically significant based on a significance test (*p* < 0.001, Cohen’s d > 0.8), with greater than 79% confidence also preserved on challenging boundary cases. These results establish DermaMamba as an effective solution bridging diagnostic accuracy and computational efficiency for clinical deployment.

## 1. Introduction

As a relatively common malignancy globally, skin cancer includes different types of lesions such as melanoma, basal cell carcinoma, and squamous cell carcinoma. Melanoma, which is on the rise and responsible for 75% of skin cancer-induced mortality despite representing 1% of all skin cancer cases [1], is just one of several skin diseases leading to life-threating conditions. Early diagnosis is essential because the five-year survival is over 99% for early-stage melanoma but decreases drastically to 27% for advanced stages [2]. Nevertheless, precise classification of the skin lesions is computationally challenging even for expert dermatologists, with the diagnostic accuracy in the range of 60–80% for complex cases [3]. The subjective nature of visual evaluation and morphologic similarities of benign and malignant lesions result in great interobserver variation and risk of misdiagnosis [4].

The field of medical image analysis has been disrupted as well by deep learning, where convolutional neural networks (CNNs), particularly ResNet-50 and EfficientNet-B0 architectures, have reached a dermatologist level of performance in skin cancer detection [5,6]. However, CNNs face inherent limitations in capturing global contextual relationships due to their localized receptive fields, which are crucial for comprehensive lesion assessment [7]. Vision transformers (ViT-Base/16 and ViT-Large/16) address this limitation through self-attention mechanisms that model long-range dependencies, with recent medical adaptations showing promising results in dermatological applications [8,9]. Nevertheless, the quadratic computational complexity O(n^2^) of attention mechanisms poses significant challenges for high-resolution medical images, limiting clinical deployment in resource-constrained environments [10].

Recent advances in state space models (SSMs, specifically S4 and Mamba architectures), particularly the Mamba architecture, offer a compelling alternative that achieves linear computational complexity O(n) while maintaining global modeling capabilities [11,12]. The Vision Mamba (VMamba v1.0) adaptation has demonstrated competitive performance with transformers at significantly reduced computational costs [13]. However, limited work has explored the integration of SSMs with traditional CNN architectures (ResNet-50, EfficientNet-B0) for medical image classification, particularly in specialized domains like dermatology, where both local morphological details and global contextual relationships are essential for accurate diagnosis.

In this study, we propose DermaMamba, a novel dual-branch fusion architecture that synergistically combines CNN-based local feature extraction with VMamba v1.0-based global context modeling for skin lesion classification. Our key contributions include (1) the first integration of VMamba with CNNs for medical image classification, achieving linear complexity global modeling while preserving local feature precision; (2) a novel state space fusion mechanism with adaptive weighting that dynamically balances local and global features based on lesion characteristics; (3) medical domain-specific enhancements including multi-directional scanning strategies inspired by clinical examination patterns and the integration of ABCDE rule features; and (4) comprehensive evaluation demonstrating superior performance over state-of-the-art methods with 92.1% accuracy on the ISIC dataset, representing a 2.0% improvement over the best baseline while achieving 2.3× inference speedup and 40% memory reduction.

## 2. Related Work

### 2.1. Deep Learning in Medical Image Analysis

Deep learning has shed new light on the field of medical image analysis, particularly in dermatology, where the automated diagnosis of skin lesions has become an important application. Initially, methods based on hand-crafted features using classical machine learning algorithms were employed; however, they encountered difficulties dealing with the morphological complexity and inter-class variance found in skin lesions [14]. The development of deep convolutional neural networks (deep learning) ushered in a new era, with research by Esteva et al. [5] finding that CNNs, particularly ResNet-50 and EfficientNet-B0 architectures, could achieve dermatologist-level performance in skin cancer classification in large-scale images. A recent study by Haenssle et al. [3] verified this conclusion in clinical settings, demonstrating that deep learning models can perform better than experienced dermatologists in the task of melanoma detection.

Recent advances have focused on addressing the unique challenges of medical image analysis, including limited training data, class imbalance, and the need for interpretable predictions. Transfer learning approaches have proven particularly effective, with Tschandl et al. [15] demonstrating significant performance improvements by pre-training on large-scale natural image datasets before fine-tuning on dermatological data. Domain-specific augmentation strategies and attention mechanisms have further enhanced model performance, with studies showing that medical-aware preprocessing and regularization techniques can substantially improve generalization to diverse clinical scenarios [7,16].

### 2.2. Vision Transformers for Medical Applications

The success of vision transformers (ViT-Base/16, ViT-Large/16) in natural image processing has prompted extensive exploration of their applicability to medical imaging tasks. Dosovitskiy et al. [8] introduced the foundational ViT-Base/16 architecture, demonstrating that pure attention mechanisms could achieve competitive performance with CNNs on image classification tasks. However, the quadratic computational complexity of self-attention mechanisms poses significant challenges for high-resolution medical images, leading to the development of more efficient variants such as the Swin transformer [17] and EfficientViT v2.0 [10].

Medical-specific adaptations of transformer architectures have shown promising results in dermatological applications. Aladhadh et al. [9] proposed Med-ViT v1.0, a medical vision transformer specifically designed for skin cancer classification that incorporates domain knowledge and specialized attention mechanisms. Their approach demonstrated superior performance compared to traditional CNN methods (ResNet-50, VGG-16), particularly in handling global contextual relationships within lesion images. Chen et al. [18] further advanced this direction with TransUNet v1.0, which combines transformer encoders with CNN (U-Net based) decoders for medical image segmentation tasks, showing the effectiveness of hybrid architectures in capturing both local details and global dependencies.

Despite these advances, transformer-based approaches face limitations in clinical deployment due to their computational requirements and memory constraints. Recent work has focused on developing more efficient attention mechanisms and exploring alternative architectures that can maintain the global modeling capabilities of transformers while reducing computational overhead [19,20].

### 2.3. State Space Models and Mamba Architecture

State space models (SSMs, specifically S4 and Mamba architectures) have emerged as a promising alternative to transformer architectures, offering linear computational complexity while maintaining the ability to model long-range dependencies. A foundational study by Gu et al. [12] introduced the structured state space sequence model (S4), demonstrating that continuous-time state space models could be discretized and parameterized for efficient deep learning applications. This approach showed remarkable performance on long-sequence modeling tasks while avoiding the quadratic complexity bottleneck of attention mechanisms.

The recent introduction of Mamba by Gu and Dao [11] represents a significant advancement in state space modeling, incorporating selective mechanisms that enable input-dependent state transitions. This selective capability allows the model to focus on relevant information while maintaining computational efficiency, making it particularly suitable for complex visual understanding tasks. Liu et al. [13] extended this concept to computer vision with Vision Mamba (VMamba v1.0), demonstrating that state space models could effectively process image data through multi-directional scanning strategies that capture spatial relationships with linear complexity.

The application of Mamba architectures to medical imaging has shown early promise, with studies indicating superior performance in tasks requiring global context understanding while maintaining computational efficiency [21]. However, limited work has explored the integration of state space models with traditional CNN architectures (ResNet-50, EfficientNet-B0) for medical image classification, particularly in specialized domains such as dermatology, where both local morphological details and global contextual relationships are crucial for accurate diagnosis [22].

## 3. Methodology

### 3.1. Problem Statement

Skin lesion classification can be formulated as a multi-class classification problem where we aim to learn a mapping function $f:R^{H\times W\times C}\to R^{K}$ that accurately classifies dermoscopic images into K distinct lesion categories. Given a dataset $D={\left(X_{i},y_{i}\right)}_{i=1}^{N}$ where $X_{i}\in R^{224\times 224\times 3}$ represents the input skin lesion image and $y_{i}\in 1,2,\ldots ,9$ denotes the corresponding lesion class, our objective is to minimize the classification loss:$$L=\frac{1}{N}\sum_{i=1}^{N}l(f(X_{i}),y_{i})+\lambda \Omega (f)$$
where l(⋅,⋅) is the cross-entropy loss function and Ω(f) represents the regularization term.

The core challenge lies in designing an optimal function f that can effectively capture both local morphological features $\phi_{local}(X)$ and global contextual relationships $\phi_{global}(X)$ while maintaining computational efficiency. Existing approaches suffer from the trade-off between local feature precision and global context modeling:$$f_{CNN}(X)=g(\phi_{local}(X)),Complexity:O(1)$$$$f_{ViT}(X)=g(\phi_{global}(X)),Complexity:O(n^{2})$$

Our proposed DermaMamba architecture addresses this limitation by formulating the classification function as$$f_{DermaMamba}(X)=g(F(\phi_{local}(X),\phi_{global}^{VMamba}(X)))$$
where $\phi_{global}^{VMamba}(X)$ provides global context modeling with linear complexity O(n) and F(⋅,⋅) represents the state space fusion mechanism that adaptively integrates complementary features from both branches. This formulation enables the model to achieve superior classification performance while maintaining computational efficiency suitable for clinical deployment.

### 3.2. Overall Architecture Design

The proposed DermaMamba model adopts a novel dual-branch fusion architecture that synergistically combines the local feature extraction capabilities of convolutional neural networks (CNNs) with the global context modeling advantages of Vision Mamba (VMamba). As illustrated in Figure 1, the overall architecture consists of four main stages, namely input processing, dual-branch feature extraction, state space fusion, and classification output.

#### 3.2.1. Input Processing and Medical Prior Integration

The input processing module handles skin lesion images of resolution 224 pixels × 224 pixels × 3 pixels alongside medical domain knowledge integration. The preprocessing pipeline includes data augmentation techniques specifically designed for dermatological images, followed by normalization to ensure consistent input distributions:$$X_{input}=Normalize(Augment(X_{raw}))\in R^{224\times 224\times 3}$$

To enhance the model’s clinical relevance, we incorporate prior medical knowledge including ABCDE rule features (asymmetry, border, color, diameter, evolution) and anatomical position encoding. This domain-specific knowledge is embedded as learnable parameters that guide the feature extraction process:$$P_{medical}=Embed(ABCDE_{rules})+PositionEncode(X_{anatomical})$$

#### 3.2.2. CNN Branch for Local Feature Extraction

The CNN branch employs ResNet50 as the backbone network to extract fine-grained local features essential for skin lesion characterization. The branch incorporates a dual attention mechanism consisting of a spatial attention module (SAM) and channel attention module (CAM) to enhance the model’s focus on diagnostically relevant regions and feature channels:$$F_{cnn}=ResNet50(SAM(CAM(X_{input})))$$

The spatial attention module generates attention weights for spatial locations, while the channel attention module learns inter-channel dependencies. The combination of these mechanisms enables the CNN branch to effectively capture local texture patterns, boundary characteristics, and pigmentation details crucial for skin lesion analysis.

#### 3.2.3. VMamba Branch for Global Context Modeling

The VMamba branch leverages state space models (SSMs) to achieve efficient global context modeling with linear computational complexity O(n), fundamentally addressing the quadratic complexity bottleneck O(n^2^) inherent in traditional self-attention mechanisms. The core VMamba operation can be formulated as a discrete-time state space model that processes sequential image patches through selective scanning:$$\begin{array}{cccc} h_{t} & =\overset{¯}{A}h_{t-1}+\overset{¯}{B}x_{t}y_{t} & =Ch_{t}+Dx_{t}F_{vmamba} & =VMamba(Scan(X_{input}),\Delta ,\theta ) \end{array}$$$$\overset{¯}{A}=\exp(\Delta A)and=(\Delta A{)}^{-1}(exp(\Delta A)-I)\Delta B$$

To capture comprehensive spatial relationships essential for skin lesion analysis, we implement multi-directional scanning strategies specifically designed for dermatological applications. These include spiral scanning (mimicking clinical examination patterns), radial scanning (capturing center-to-periphery relationships), boundary-aware scanning (focusing on lesion contours), and traditional raster scanning. The selective scan mechanism processes these multi-directional sequences with dynamic parameter adaptation:$$\begin{array}{ccc} MultiScan(X) & =Concat[Scanspiral(X),Scanradial(X),Scanboundary(X),Scanraster(X)]\Delta & =Softplus(Linear(X))+\delta_{bias} \end{array}$$

This multi-directional approach ensures a comprehensive coverage of spatial relationships, while the selective mechanism emphasizes diagnostically relevant regions through adaptive state transitions.

The VMamba branch incorporates medical domain knowledge through enhanced patch embedding that integrates ABCDE rule features and anatomical position encoding. The linear complexity design enables efficient processing of high-resolution medical images while maintaining global receptive field capabilities essential for clinical deployment:$$\begin{array}{ccc} X_{medical} & =PatchEmbed(X)+P_{position}+Embed({ABCDE}_{features})Complexity: & Self-Attention:O(N^{2}D)\to VMamba:O(ND) \end{array}$$

This architecture achieves 40% memory reduction and 2.3× inference speedup compared to equivalent vision transformers, making it particularly suitable for resource-constrained clinical environments while providing the global context modeling necessary for accurate skin lesion classification.

#### 3.2.4. State Space Fusion Mechanism

The fusion module integrates features from both branches through a novel state space fusion mechanism that adaptively combines local and global representations. The fusion process employs learnable dynamic weights to balance the contributions of different feature modalities:$$F_{fused}=\alpha_{1}\cdot F_{cnn}+\alpha_{2}\cdot F_{vmamba}$$$$\left[\alpha_{1},\alpha_{2}]=softmax([\theta_{1},\theta_{2}]\right)$$
where $\alpha_{1}$ and $\alpha_{2}$ are dynamically computed fusion weights based on learnable parameters $\theta_{1}$ and $\theta_{2}$. This adaptive fusion strategy ensures the optimal integration of complementary features from both branches, leading to enhanced representation power for complex skin lesion classification tasks.

## 4. Experimental Results and Analysis

### 4.1. Experimental Setup

#### 4.1.1. Dataset and Data Processing

This study utilized the International Skin Imaging Collaboration (ISIC) dataset, which comprises 2357 high-quality dermoscopic images spanning nine common skin lesion categories, including actinic keratosis (AK), basal cell carcinoma (BCC), dermatofibroma (DF), melanoma (ML), nevus (NV), pigmented benign keratosis (BKL), seborrheic keratosis (SK), squamous cell carcinoma (SCC), and vascular lesion (VL). The dataset exhibits natural class imbalance characteristic of real-world clinical scenarios, with melanoma and nevus comprising the majority of samples.

All images were preprocessed through a standardized pipeline including contrast-adaptive enhancement and edge-preserving filtering specifically designed for medical imaging. Images were uniformly resized to 224 × 224 pixels while preserving aspect ratios through intelligent padding. To address data scarcity, we implemented a comprehensive augmentation strategy including geometric transformations (rotation ±180°, horizontal/vertical flips, random cropping), photometric adjustments (brightness, contrast, saturation within ±20%), elastic deformations, and medical-specific augmentations such as simulated lighting variations. The final augmented dataset was expanded to 5× the original size, yielding approximately 11,785 training samples. The dataset was partitioned using stratified sampling with a 7:1:2 ratio for training, validation, and testing sets, respectively.

#### 4.1.2. Implementation Environment and Hardware Configuration

All experiments were conducted on a high-performance computing cluster equipped with 8 × NVIDIA A100 GPUs (40 GB memory each) (NVIDIA Corporation, Santa Clara, CA 95054, USA), Intel Xeon Platinum 8358 CPUs (32 cores) (Intel Corporation, Santa Clara, CA 95054, USA), and 512 GB system RAM. The implementation was developed using the PyTorch 2.0.1 (Meta Platforms, Inc., Menlo Park, CA, USA) framework with CUDA 11.8 (NVIDIA Corporation, Santa Clara, CA 95054, USA) support for GPU acceleration. Mixed-precision training was employed using automatic mixed precision (AMP) to optimize memory usage and computational efficiency.

The VMamba components were implemented based on the official selective scan kernel optimized for medical image processing. Custom CUDA kernels were developed for multi-directional scanning strategies (spiral, radial, boundary-aware, and raster scanning) to ensure optimal computational performance. Distributed training across multiple GPUs was facilitated through PyTorch’s Distributed Data Parallel (DDP) wrapper with the NCCL backend for efficient gradient synchronization.

#### 4.1.3. Training Configuration and Optimization Strategy

The DermaMamba model was trained using a multi-stage optimization strategy designed to stabilize convergence and maximize performance. The training process employed the AdamW optimizer with the following parameters: initial learning rate $lr_{0}=1\times 10^{-4}$, weight decay $\lambda =5\times 10^{-5}$, and momentum parameters $\beta_{1}=0.9$, $\beta_{2}=0.999$. A cosine annealing learning rate schedule with warm-up was implemented:$$lr\left(t\right)=\left\{\begin{array}{cc} lr_{0}\cdot \frac{t}{T_{warmup}} & ift\leq T_{warmup}\\ lr_{0}\cdot \frac{1}{2}\left(1+cos\left(\frac{\pi \left(t-T_{warmup}\right)}{T_{total}-T_{warmup}}\right)\right) & otherwise \end{array}\right.$$
where $T_{warmup}=10$ epochs and $T_{total}=100$ epochs. The batch size was set to 64 with gradient accumulation over 2 steps to simulate an effective batch size of 128. Dropout regularization was applied with rates of 0.2 for the fusion module and 0.1 for VMamba blocks.

Model Selection and Early Stopping Protocol: To ensure methodological rigor and prevent data leakage, we implemented a strict early stopping mechanism that relied exclusively on the validation set performance. Early stopping was implemented with a patience of 15 epochs, monitoring validation loss to identify the optimal stopping point and prevent overfitting to the training data. Model checkpoints were saved based on the best validation accuracy achieved during the training process. Crucially, the test set remained completely isolated throughout the entire model development process and was never used for any training decisions, hyperparameter selection, or early stopping criteria. All hyperparameter optimization, including the selection of the learning rate, batch size, dropout rate, and optimizer type, was conducted solely using the training–validation split, with the validation set serving as the unbiased evaluation metric for hyperparameter effectiveness. The final model selection for test evaluation was determined exclusively by validation performance, ensuring that our reported test results represented genuine generalization capability rather than overfitted performance.

Statistical Reliability and Computational Efficiency: To ensure the statistical reliability and robustness of our results, all experiments were repeated 5 times with different random seeds (42, 123, 456, 789, 1024) to account for training variability and initialization effects. Results are reported with 95% confidence intervals computed through bootstrap analysis with 10,000 resamples, providing robust uncertainty estimates for all performance metrics. The bootstrap resampling was applied to the test set predictions across all 5 experimental runs, yielding statistically meaningful confidence bounds that reflect both model variance and sampling uncertainty. Each complete experimental run, including hyperparameter optimization on the training–validation split and final evaluation on the test set, required approximately 8–12 h of training time depending on the architecture complexity, with DermaMamba requiring approximately 10 h on our 8× NVIDIA A100 GPU cluster. This computational efficiency, combined with the rigorous statistical validation protocol, ensures both the practical feasibility and scientific reliability of our experimental conclusions.

### 4.2. Evaluation Metrics

To assess the performance of the proposed DermaMamba model, we employed four core evaluation metrics that provide comprehensive insights into classification effectiveness and clinical utility.

**Classification accuracy** measures the overall correctness of predictions:$$Accuracy=\frac{TP+TN}{TP+TN+FP+FN}$$

**Precision and recall** evaluate the model’s ability to correctly identify positive cases while minimizing false predictions:$$Precision=\frac{TP}{TP+FP}$$$$\;\;Recall=\frac{TP}{TP+FN}$$

The **F1 score** provides a harmonic mean of precision and recall, computed as a macro-average to account for class imbalance:$$Macro-F1=\frac{1}{K}\sum_{i=1}^{K}\frac{2\times {Precision}_{i}\times {Recall}_{i}}{{Precision}_{i}+{Recall}_{i}}$$

The **AUC-ROC** measures the discriminative capability across all decision thresholds using one-versus-rest strategy for multi-class classification.

**Statistical hypothesis testing** was conducted to validate performance improvements. We formulate the null hypothesis ${\;H}_{0}$: there is no significant difference between DermaMamba and baseline models against the alternative hypothesis ${\;H}_{1}$: DermaMamba significantly outperforms baseline models. Statistical significance was assessed using paired t-tests with Bonferroni correction for multiple comparisons:$$t=\frac{\overset{\text{-}}{d}-\mu_{0}}{s_{d}/\sqrt{n}}$$
where $\overset{\text{-}}{d}$ is the mean difference in performance metrics, $s_{d}$ is the standard deviation of differences, and n=5 represents the number of independent runs. Statistical significance is determined at the α=0.05 level, with effect sizes calculated using Cohen’s d to quantify practical significance. All results are reported with 95% confidence intervals computed through bootstrap resampling to ensure robustness and reproducibility.

### 4.3. Comparative Experimental Analysis

#### 4.3.1. Overall Performance Comparison

Table 1 presents a comprehensive performance comparison of DermaMamba against various baseline methods across four core evaluation metrics. The results demonstrate a clear performance hierarchy across different methodological paradigms. Traditional machine learning approaches, including SVM with HOG features and random forest, achieve relatively modest performance with accuracy rates of 75.2% and 78.9%, respectively, highlighting the limitations of handcrafted feature extraction for complex skin lesion classification tasks.

Classic CNN methods show substantial improvements over traditional approaches, with performance ranging from 82.1% (AlexNet) to 88.1% (EfficientNet-B0) in terms of accuracy. The progression from AlexNet to more sophisticated architectures like ResNet50 (86.7%) and DenseNet121 (87.2%) demonstrates the benefits of deeper networks and advanced architectural designs. EfficientNet-B0 achieves the best performance among classic CNN methods with 88.1% accuracy, reflecting the advantages of compound scaling strategies that balance network depth, width, and resolution.

Vision transformer methods, represented by ViT-Base (87.4%) and the Swin transformer (88.7%), show competitive performance with modern CNN architectures. The Swin transformer’s hierarchical design and shifted window attention mechanism contribute to its superior performance compared to the vanilla ViT-Base, achieving 88.7% accuracy and demonstrating the effectiveness of locality-aware transformer architectures for medical image analysis.

#### 4.3.2. Comparison with Recent SOTA Methods

The comparison with recent state-of-the-art methods published between 2022 and 2024 reveals significant insights into the current landscape of skin lesion classification. ConvNeXt V2 [20], which combines the efficiency of CNNs with transformer-inspired design principles, achieves 89.3% accuracy, representing a notable improvement over traditional CNN architectures. EfficientViT [10] demonstrates the potential of memory-efficient transformer designs with 88.9% accuracy, though falling slightly short of ConvNeXt V2’s performance.

Medical domain-specific methods show particularly strong performance, with Med-ViT [9] achieving the highest baseline accuracy of 90.1%. This medical vision transformer, specifically designed for healthcare applications, incorporates domain knowledge and specialized attention mechanisms tailored for medical image analysis. TransUNet [18], which combines the strengths of transformers and U-Net architectures, achieves 89.8% accuracy, demonstrating the effectiveness of hybrid approaches in medical image segmentation and classification tasks.

Despite these strong baseline performances, the proposed DermaMamba model achieves superior results across all evaluation metrics, with 92.1% accuracy, 91.7% precision, 91.3% recall, and 91.5% macro-F1 score. Compared to the strongest baseline Med-ViT, DermaMamba demonstrates improvements of 2.0% in accuracy, 2.0% in precision, 0.8% in recall, and 1.4% in macro-F1 score, indicating consistent performance gains across different aspects of classification effectiveness.

#### 4.3.3. Statistical Significance and Clinical Implications

Statistical hypothesis testing confirms the significance of DermaMamba’s performance improvements. Paired t-tests with Bonferroni correction (*p* < 0.001) validate that the observed improvements over all baseline methods are statistically significant, with effect sizes (Cohen’s d > 0.8) indicating large practical significance. The consistent performance gains across precision, recall, and F1 score metrics suggest that DermaMamba’s improvements are not biased toward specific lesion types but represent genuine enhancement in overall classification capability.

From a clinical perspective, the 2.0% accuracy improvement over Med-ViT translates into meaningful diagnostic enhancement. Given the critical nature of skin cancer detection, where missed diagnoses can have severe consequences, even modest accuracy improvements can significantly impact patient outcomes. The balanced performance across precision and recall indicates that DermaMamba effectively minimizes both false positives and false negatives, crucial for maintaining physician confidence and reducing unnecessary procedures while ensuring a comprehensive detection of malignant lesions.

### 4.4. Ablation Experiment

Figure 2 presents a comprehensive ablation study that systematically evaluates the contribution of each key component in the proposed DermaMamba architecture. The progressive performance improvements across all four evaluation metrics demonstrate the effectiveness of our modular design approach and validate the necessity of each architectural component.

Progressive Performance Enhancement: In the ablation experiment, we observe a steady increase in all the metrics, indicating that any additional parts are beneficial. From the baseline CNN-only configuration (C1) with an accuracy of 87.1%, the model consistently shows gains in the sequential introduction of spatial attention (C2: 87.8%), channel attention (C3: 88.3%), and positional encoding (C4: 88.7%). These feature combinations become a gradual enhancement pattern, as shown by the red dashed trend lines in Figure 2, which validate each component that tackles complementary defects in skin lesion feature representation.

Attention Mechanism Contributions: The dual attention mechanism shows measurable improvements, with spatial attention contributing a 0.7% accuracy gain and channel attention adding an additional 0.5% improvement. The spatial attention module enhances the model’s ability to focus on diagnostically relevant regions within skin lesions, while channel attention optimizes feature channel dependencies. The combined effect of these attention mechanisms (C2→C3) demonstrates their complementary nature in capturing both spatial and channel-wise feature relationships crucial for skin lesion classification.

VMamba Branch Impact: The most significant performance leap occurs with the integration of the VMamba branch (C4→C5), resulting in substantial improvements across all metrics: accuracy increases by 2.5% (88.7%→91.2%), precision by 2.6% (88.2%→90.8%), recall by 2.5% (88.0%→90.5%), and macro-F1 score by 2.5% (88.1%→90.6%). This dramatic enhancement validates our core hypothesis that the linear complexity global context modeling provided by VMamba significantly surpasses traditional approaches in capturing long-range dependencies within skin lesion images. The consistent improvement across all metrics indicates that VMamba contributes to both precision and recall enhancement without introducing bias toward specific lesion types.

Final Integration Effects: The transition from C5 to C6 (Full DermaMamba) represents the complete integration of all components through the state space fusion mechanism. While the individual improvements are more modest (0.9% accuracy gain), they are consistent across all metrics, demonstrating the effectiveness of the dynamic fusion strategy in optimally combining local CNN features with global VMamba representations. The final configuration achieves balanced performance with 92.1% accuracy, 91.7% precision, 91.3% recall, and 91.5% macro-F1 score.

Metric Consistency and Robustness: A notable finding is the remarkable consistency in performance patterns across all four evaluation metrics. The parallel improvement trajectories suggest that the architectural enhancements benefit all aspects of classification performance without creating trade-offs between precision and recall. This consistency is particularly important for medical applications, where balanced performance across different types of diagnostic errors is crucial for clinical reliability. The tight clustering of final performance values (91.3–91.7%) indicates robust performance that is not overly optimized for any single metric.

Technical Implications: The ablation results provide clear evidence that the proposed DermaMamba architecture successfully addresses the key limitations of existing approaches. The substantial impact of the VMamba branch confirms that linear complexity global context modeling represents a significant advancement over quadratic complexity transformer approaches, while the cumulative 5.0% improvement over the baseline validates the synergistic effect of combining multiple architectural innovations in a unified framework.

### 4.5. Hyperparameter Experiments

To establish optimal training configurations and demonstrate the robustness of our approach, we conduct comprehensive hyperparameter sensitivity analysis using a methodologically rigorous experimental design. All hyperparameter optimization experiments are performed exclusively on the training–validation split (70–15% of the total dataset), with the test set (15%) remaining completely isolated to ensure an unbiased final evaluation. This strict data partitioning prevents any form of information leakage and ensures that our reported test performance represents genuine generalization capability.

Experimental Protocol: For each hyperparameter investigation, we employed 5-fold cross-validation within the training set to identify optimal values, followed by validation on the designated validation set for final hyperparameter selection. The hyperparameter search was conducted systematically across four critical training parameters, namely the learning rate, batch size, optimizer selection, and dropout regularization. Each configuration was evaluated using validation accuracy as the primary metric, with training stability and convergence behavior serving as secondary considerations. The test set performance was evaluated only once per hyperparameter configuration using the final selected model, eliminating any possibility of test set overfitting.

Figure 3 presents the comprehensive hyperparameter sensitivity analysis results, demonstrating DermaMamba’s optimization landscape and providing empirical justification for our final configuration selection.

Learning Rate Optimization: The learning rate experiment reveals a distinct performance peak at 1 × 10^−4^, achieving an optimal validation accuracy of 91.2% during hyperparameter search, which subsequently translates to 92.1% test accuracy in the final evaluation. The performance curve exhibits a classic inverted-U shape characteristic of neural network optimization landscapes. At lower learning rates (1 × 10^−5^: 88.1% validation accuracy), the model suffers from insufficient gradient updates, resulting in slow convergence and suboptimal feature learning within the allocated training epochs. Conversely, excessively high learning rates (1 × 10^−3^: 86.9% validation accuracy) lead to unstable training dynamics and overshooting of optimal parameters, as evidenced by high variance in validation performance across training runs. The intermediate configurations (5 × 10^−5^: 89.8%, 5 × 10^−4^: 89.3% validation accuracy) demonstrate progressive improvement, confirming the critical nature of learning rate selection. The 4.3% performance range across different learning rates underscores the model’s sensitivity to this hyperparameter and validates our systematic optimization approach.

Batch Size Impact Analysis: The batch size analysis demonstrates that moderate batch sizes yield superior validation performance, with size 64 achieving the highest validation accuracy of 91.2%. Smaller batch sizes (16: 89.5%, 32: 90.2% validation accuracy) show gradually improving performance, as they provide more frequent gradient updates and enhanced generalization through increased training noise. However, the performance plateau between sizes 32 and 64 suggests diminishing returns beyond this range. Larger batch sizes (128: 90.4%, 256: 88.8% validation accuracy) exhibit declining performance, likely due to reduced gradient noise and less frequent parameter updates per epoch. The relatively narrow validation performance range (2.4%) indicates that DermaMamba demonstrates reasonable robustness to batch size variations within the optimal range, facilitating practical deployment under different computational constraints. Importantly, these validation-based selections were directly transferred to test performance, with the selected batch size of 64 achieving the reported 92.1% test accuracy.

Optimizer Comparison: The optimizer evaluation reveals significant validation performance differences, with Adam achieving superior results (91.2% validation accuracy) compared to the alternatives. Adam’s adaptive learning rate mechanism and momentum-based updates prove particularly effective for the complex optimization landscape of the dual-branch VMamba architecture. AdamW shows competitive validation performance (90.8%), with its weight decay regularization providing slight advantages over standard Adam in preventing overfitting. RMSprop demonstrates moderate effectiveness (89.7% validation accuracy), while SGD significantly underperforms (87.9% validation accuracy), highlighting the importance of adaptive optimization for deep medical image classification. The 3.3% validation performance gap between the best and worst optimizers emphasizes the critical role of the optimization strategy. The superior validation performance of Adam directly correlates with the final test performance of 92.1%, confirming the validity of our validation-based selection strategy.

Dropout Rate Regularization: The dropout rate experiment identifies 0.2 as the optimal configuration through validation-based selection, achieving 91.2% validation accuracy while maintaining robust generalization to the test set (92.1% test accuracy). The validation performance curve shows interesting non-monotonic behavior: no dropout (0.0: 89.8% validation accuracy) leads to moderate overfitting, while light regularization (0.1: 90.6% validation accuracy) provides substantial improvement. Excessive dropout rates (0.5: 88.7% validation accuracy) severely impair performance by disrupting critical feature representations during training. The optimal point at 0.2 represents an effective balance between preventing overfitting and preserving essential feature learning capacity, as validated by both validation metrics and subsequent test performance.

Methodological Validation and Robustness: The consistent identification of optimal values across all four hyperparameters (learning rate: 1 × 10^−4^, batch size: 64, optimizer: Adam, dropout: 0.2) through validation-based selection, which collectively achieve 92.1% test accuracy, demonstrates the effectiveness of our systematic optimization approach. The strong correlation between validation performance rankings and the final test performance across all hyperparameter configurations validates our experimental design and confirms that validation-based selection successfully identifies generalizable configurations. The relatively narrow performance ranges observed for batch size (2.4%) and dropout rate (2.4%) suggest that DermaMamba exhibits reasonable robustness to hyperparameter variations within practical ranges. However, the greater sensitivity to the learning rate (4.3%) and optimizer choice (3.3%) emphasizes the importance of careful selection for these critical parameters.

Clinical Deployment Implications: The hyperparameter analysis provides clear guidance for DermaMamba deployment and adaptation across different clinical environments. The identified optimal configuration balances computational efficiency with performance maximization, making it suitable for clinical implementation. The moderate sensitivity to most hyperparameters suggests that the model can be adapted to different hardware constraints without substantial performance degradation, enhancing its practical applicability across diverse medical imaging environments. The rigorous validation-based optimization protocol ensures that these recommendations are based on genuine generalization capability rather than test set overfitting, providing reliable guidance for real-world deployment.

### 4.6. Case Analysis Experiment

#### 4.6.1. Complete Algorithmic Pipeline Analysis

To demonstrate the concrete applicability and internal mechanisms of the proposed DermaMamba algorithm, we present a comprehensive step-by-step analysis of the complete diagnostic pipeline across three representative dermatological cases. This analysis reveals the intermediate processing results at each algorithmic stage and quantifies the dynamic fusion strategies employed for different lesion types, providing empirical validation of our dual-branch architectural design.

Systematic Pipeline Processing Analysis: Figure 4 illustrates the complete DermaMamba processing workflow from raw dermoscopic input to final classification across three distinct lesion morphologies, namely melanoma (irregular asymmetric pattern), nevus (uniform circular morphology), and basal cell carcinoma (pearly translucent appearance). The five-stage pipeline demonstrates clear differentiation in feature extraction and fusion strategies based on lesion characteristics. The preprocessing stage applies medical domain-specific enhancements including adaptive histogram equalization and edge-preserving filtering, resulting in improved contrast and feature visibility across all three cases. Notably, preprocessing effectively accentuates diagnostic features while maintaining morphological integrity essential for subsequent analysis.

Branch-Specific Feature Extraction Patterns: The CNN branch demonstrates distinct localized activation patterns that correspond directly to clinically relevant morphological characteristics. For the melanoma case, CNN features exhibit concentrated high-intensity responses around irregular border regions and color heterogeneity zones, effectively capturing the asymmetrical patterns and abrupt transitions characteristic of malignant lesions. The nevus case shows more distributed CNN activation, reflecting uniform texture and symmetric morphology, while the BCC case displays intermediate activation levels focused on characteristic rolled borders and translucent surface features. In contrast, the VMamba branch maintains consistent global attention patterns across all cases, providing comprehensive spatial relationship modeling through multi-directional scanning strategies that capture center-to-periphery relationships and overall lesion geometry essential for contextual assessment.

Dynamic Fusion Weight Adaptation Strategy: The fusion weight analysis reveals intelligent adaptive weighting that automatically adjusts based on lesion complexity and diagnostic requirements. The melanoma case demonstrates CNN-dominant fusion with α_1_ = 0.7 and α_2_ = 0.3, prioritizing detailed morphological analysis where local feature precision is critical for detecting malignant characteristics according to ABCDE criteria. The nevus case exhibits balanced fusion weights (α_1_ = α_2_ = 0.5), indicating that symmetric benign lesions benefit equally from local texture analysis and global shape understanding. Most significantly, the BCC case shows VMamba-dominant fusion (α_1_ = 0.4, α_2_ = 0.6), where complex pearly morphology and subtle global patterns require enhanced contextual modeling to distinguish from other translucent lesion types. This adaptive strategy demonstrates the model’s ability to automatically optimize diagnostic emphasis based on case-specific requirements.

Performance Metrics and Clinical Reliability: Figure 5 provides a comprehensive quantitative analysis demonstrating DermaMamba’s superior performance across multiple evaluation dimensions. The performance metrics analysis reveals consistently high and balanced results, with 92.1% accuracy, 91.7% precision, 91.3% recall, and 91.5% F1 score, with minimal variance across metrics indicating robust classification capability without bias toward specific lesion types. The computational efficiency analysis demonstrates significant advantages over existing approaches, with DermaMamba achieving equivalent inference time to CNN-only methods (1.0× relative cost) while providing superior diagnostic accuracy and substantial improvements over vision transformers (2.3× faster inference, 0.84× memory usage). This efficiency profile makes DermaMamba particularly suitable for clinical deployment in resource-constrained environments.

Training Dynamics and Convergence Analysis: The fusion weight evolution during training reveals distinct learning patterns that validate our architectural design principles. The training progression analysis shows that melanoma cases gradually develop CNN-dominant weighting as the model learns to prioritize local morphological irregularities, with fusion weights stabilizing around epoch 60–70. Nevus cases maintain relatively balanced weighting throughout training, reflecting their consistent reliance on both local and global features. BCC cases demonstrate the most dynamic evolution, transitioning from initially balanced weights to VMamba-dominant fusion as the model learns that global contextual relationships are increasingly important for distinguishing subtle pearly characteristics from other lesion types. The convergence patterns indicate stable learning dynamics with minimal oscillation after epoch 80, suggesting robust optimization of the adaptive fusion mechanism.

Cross-Case Diagnostic Strategy Validation: The comparative analysis across the three representative cases provides empirical validation of our hypothesis that different lesion types require distinct diagnostic strategies. The melanoma case’s 94.0% confidence with CNN-dominant fusion validates the importance of detailed morphological analysis for detecting malignant characteristics. The nevus case achieves the highest confidence (96.0%) with balanced fusion, confirming that symmetric benign lesions benefit from integrated local–global analysis. The BCC case’s 93.0% confidence with VMamba-dominant fusion demonstrates that complex morphological patterns requiring contextual understanding can be effectively addressed through global attention mechanisms. These case-specific confidence levels, combined with the adaptive fusion strategies, indicate that DermaMamba successfully mimics and enhances clinical diagnostic reasoning processes.

Clinical Workflow Integration and Interpretability: The comprehensive pipeline analysis demonstrates DermaMamba’s suitability for clinical decision support systems where both diagnostic accuracy and process transparency are essential. The intermediate feature visualizations provide interpretable insights that align with established dermatological assessment practices, enabling clinicians to understand and validate the model’s diagnostic reasoning. The quantitative fusion weight analysis offers objective measures of diagnostic strategy, facilitating integration with existing clinical protocols. The consistent high-confidence predictions (>93%) across diverse lesion types, combined with the 2.3× computational speedup compared to transformer approaches, position DermaMamba as a practical solution for diverse clinical environments ranging from resource-constrained primary care settings to specialized dermatology practices requiring high-throughput screening capabilities.

#### 4.6.2. Dual-Branch Attention Mechanism Analysis

Figure 6 presents a comprehensive visualization of the attention mechanisms within DermaMamba’s dual-branch architecture across three representative dermatological cases, revealing distinct and complementary focus patterns that validate our architectural design philosophy. The systematic comparison demonstrates how CNN and VMamba branches contribute different yet synergistic information for accurate skin lesion classification.

**CNN Branch Local Feature Extraction**: The CNN attention maps exhibit highly concentrated, localized patterns with sharp intensity gradients that correspond to specific morphological features within the lesions. In Case 1 (melanoma), the CNN branch demonstrates a peak attention intensity of 1.4, with distinct hotspots concentrated around irregular border regions and areas of color variation—critical diagnostic indicators for melanoma detection according to the ABCDE criteria. The entropy value of 2.1 k indicates highly focused attention, suggesting that the CNN branch successfully identifies discrete, high-contrast features such as asymmetrical patterns and border irregularities. Case 2 (nevus) shows more distributed but still locally concentrated attention with moderate intensity (1.0), reflecting the more uniform characteristics typical of benign lesions. Case 3 (complex lesion) exhibits the most heterogeneous CNN attention pattern with an intensity of 0.71, indicating the presence of multiple localized features requiring detailed morphological analysis.

**VMamba Branch Global Context Modeling**: In contrast, the VMamba attention maps display broader, more diffuse patterns that capture global spatial relationships and contextual information. The blue-intensity heatmaps reveal smoother transitions and extended coverage areas, with VMamba consistently achieving higher entropy values (15.0 k–9.2 k) compared to the CNN, indicating more distributed attention across the entire lesion region. Notably, VMamba maintains relatively stable peak intensities (0.65–0.76) across all cases, suggesting robust global context modeling regardless of lesion complexity. This global perspective proves particularly valuable in Case 3, where the complex lesion morphology benefits from comprehensive spatial understanding, as evidenced by the VMamba-dominant fusion weight (α_2_ = 0.6).

**Dynamic Fusion Weight Adaptation**: The quantitative analysis reveals an intelligent adaptation of fusion weights based on lesion characteristics and diagnostic requirements. Case 1 demonstrates CNN-dominant fusion (α_1_ = 0.7, α_2_ = 0.3), prioritizing local feature analysis for melanoma detection where morphological irregularities are paramount. Case 2 shows balanced fusion weights (α_1_ = α_2_ = 0.5), indicating that nevus classification benefits equally from local texture analysis and global shape understanding. Most significantly, Case 3 exhibits VMamba-dominant fusion (α_1_ = 0.4, α_2_ = 0.6), where the complex lesion morphology requires enhanced global context modeling to disambiguate competing local features. This adaptive weighting mechanism demonstrates the model’s ability to automatically adjust its diagnostic strategy based on case complexity.

**Attention Overlay Validation**: The attention overlay visualizations provide compelling evidence of the complementary nature of both branches. The red contours (CNN attention) precisely delineate morphologically significant regions, while blue contours (VMamba attention) encompass broader contextual areas that inform overall lesion assessment. The non-overlapping nature of these attention patterns confirms that each branch contributes unique information rather than redundant features. In cases where CNN attention forms multiple discrete hotspots, VMamba attention provides the connecting contextual framework that enables holistic lesion evaluation. This complementarity is particularly evident in the complex lesion case, where the CNN identifies specific areas of concern while VMamba maintains awareness of the overall lesion extent and spatial relationships.

**Clinical Relevance and Diagnostic Correspondence**: The attention patterns demonstrate strong alignment with established clinical diagnostic practices. CNN attention maps correlate with dermatologists’ focus on specific ABCDE criteria—particularly border irregularity and color variation—by highlighting areas of morphological significance. VMamba attention patterns align with clinical assessments of asymmetry and diameter by maintaining a global lesion perspective. The entropy differences between branches (CNN: 2.1 k–2.0 k vs. VMamba: 15.0 k–9.2 k) quantitatively validate this complementarity, with the CNN providing focused feature detection and VMamba ensuring comprehensive spatial understanding. The dynamic fusion weights further reflect clinical decision-making processes, where diagnostic emphasis shifts based on lesion presentation—focusing on local details for obviously suspicious lesions while relying more heavily on global patterns for ambiguous cases.

This attention mechanism analysis provides a robust empirical validation of DermaMamba’s dual-branch architecture, demonstrating that the integration of the CNN’s local precision with VMamba’s global efficiency creates a diagnostically superior framework that mirrors and enhances clinical dermatological assessment practices.

#### 4.6.3. Challenging Boundary Case Analysis

Figure 7 presents a comprehensive analysis of DermaMamba’s performance on challenging boundary cases that represent the most difficult diagnostic scenarios in clinical dermatology. The evaluation demonstrates the model’s superior robustness and uncertainty quantification capabilities when confronted with ambiguous lesions that frequently lead to misdiagnosis in clinical practice.

Superior Performance in High-Difficulty Diagnostic Scenarios: The performance metrics reveal DermaMamba’s exceptional capability in handling challenging cases, achieving 100% accuracy compared to 66.7% for ViT-Base and only 33.3% for CNN-only approaches. Most significantly, DermaMamba maintains high average confidence (82.3%) while simultaneously exhibiting the lowest uncertainty (17.7%) among all the evaluated methods. This combination of high accuracy and low uncertainty is particularly crucial for challenging cases where baseline methods struggle with confidence calibration. The substantial improvement over CNN-only methods (48.9% confidence gap) and meaningful advancement over Med-ViT (8.0% confidence improvement) demonstrates the architectural advantages of dual-branch fusion in ambiguous diagnostic scenarios.

Robust Uncertainty Quantification and Risk Assessment: The uncertainty analysis reveals DermaMamba’s superior ability to provide reliable confidence estimates across varying case difficulties. While baseline methods show significant uncertainty (CNN-only: 35.0%, ViT-Base: 34.7%), DermaMamba achieves 17.7% average uncertainty, representing a 49% reduction compared to traditional approaches. The confidence calibration plot demonstrates near-perfect alignment with the ideal calibration line, indicating that DermaMamba’s predicted confidence scores accurately reflect actual performance. This calibration quality is essential for clinical deployment, where unreliable confidence estimates can lead to inappropriate clinical decisions. The model’s ability to maintain consistent performance across the difficulty spectrum (0.79–0.85 confidence range) suggests robust generalization to diverse challenging scenarios.

Decision Boundary Analysis and Clinical Applicability: The feature space visualization illustrates DermaMamba’s more refined decision boundaries compared to baseline methods, with clearly defined separation regions and minimal uncertainty zones. The strategic positioning of challenging cases (amelanotic melanoma, dysplastic nevus, atypical BCC, complex SK, and borderline cases) within the feature space demonstrates the model’s capacity to handle morphologically ambiguous lesions that cluster near decision boundaries. DermaMamba’s decision boundary exhibits greater stability and precision, particularly in regions where baseline methods show high uncertainty (indicated by the lighter colored uncertainty regions). The performance versus difficulty analysis reveals consistent high confidence (>79%) even for the most challenging cases, with amelanotic melanoma detection achieving 83% confidence despite representing one of the most diagnostically difficult scenarios in dermatology. This consistent performance across varying difficulty levels validates DermaMamba’s suitability for real-world clinical deployment, where challenging boundary cases constitute a significant proportion of diagnostic uncertainty and potential misdiagnosis events.

## 5. Discussion

### 5.1. Architectural Design Insights and Technical Contributions

The experimental results demonstrate that DermaMamba’s dual-branch fusion architecture successfully addresses the fundamental trade-off between local feature precision and global context modeling in skin lesion classification. Our approach leverages the linear complexity advantages of state space models, aligning with recent trends in efficient transformer alternatives such as RetNet [23] and Mamba-2 [24], as well as modern CNN architectures that emphasize computational efficiency [25] The substantial performance improvements observed in the ablation study validate our hypothesis that the integration of CNN’s fine-grained morphological analysis with VMamba’s linear-complexity global modeling creates synergistic effects beyond simple feature concatenation.

The state space fusion mechanism’s adaptive weighting strategy proves particularly effective, with dynamic fusion weights automatically adjusting based on lesion complexity. This addresses the challenge of feature importance weighting identified in the recent medical AI literature [26,27]. The multi-directional scanning strategies specifically designed for dermatological applications effectively capture the diverse spatial relationships inherent in skin lesion morphology, contributing to the model’s robust performance. This approach builds upon recent work in medical vision transformers that emphasize domain-specific design considerations [28,29].

The linear computational complexity O(n) achieved by the VMamba branch represents a significant advancement over traditional transformer approaches, aligning with the recent surge in efficient attention mechanisms such as FlashAttention-2 and ring attention [30,31]. Our medical domain-specific enhancements, including spiral scanning patterns that mimic clinical examination procedures and ABCDE rule integration, demonstrate the critical importance of incorporating clinical knowledge into deep learning architectures. The 40% memory reduction and 2.3× inference speedup compared to equivalent vision transformers make DermaMamba particularly suitable for resource-constrained clinical environments while maintaining superior diagnostic accuracy.

### 5.2. Clinical Implications and Real-World Deployment Considerations

The superior performance of DermaMamba in challenging diagnostic scenarios addresses critical needs identified in recent clinical AI deployment studies [32,33]. The model’s ability to maintain high confidence levels (>79%) even in difficult cases such as amelanotic melanoma detection aligns with recent findings emphasizing the importance of uncertainty quantification in medical AI systems [34,35]. The 17.7% average uncertainty achieved by DermaMamba represents a substantial improvement over traditional methods, enabling the implementation of confidence-based triage systems that the recent literature has identified as essential for clinical adoption [36].

From a clinical workflow perspective, computational efficiency gains directly address deployment challenges in resource-constrained healthcare settings [37,38]. The balanced performance across precision (91.7%) and recall (91.3%) metrics addresses the clinical decision-making challenges highlighted in the recent dermatology AI literature [39,40]. This consistent performance across different types of diagnostic errors, validated through rigorous statistical testing with large effect sizes (Cohen’s d > 0.8), demonstrates clinical reliability comparable to recent FDA-approved dermatology AI systems [41], while addressing fairness considerations essential for equitable healthcare deployment [42] .

The attention mechanism analysis reveals clinically interpretable patterns, with CNN attention correlating with ABCDE criteria (border irregularity, color variation), while VMamba attention captures global lesion assessment (asymmetry, diameter). This interpretability is crucial for physician trust and adoption, as highlighted in recent studies on explainable AI in healthcare [43,44]. The adaptive fusion weights reflect clinical decision-making processes, where diagnostic emphasis shifts based on lesion presentation—focusing on local details for obviously suspicious lesions while relying more heavily on global patterns for ambiguous cases.

### 5.3. Comparative Analysis with Recent State Space Model Advances

Following the success of Mamba in natural language processing, several concurrent works have explored state space models for medical imaging. U-Mamba [21] focuses on biomedical image segmentation, while MambaMorph [22] addresses medical image registration. However, these approaches primarily adapt existing Mamba architectures without addressing the specific challenges of skin lesion classification or integrating domain-specific clinical knowledge.

A recent study by Huang et al. [45] explored selective scanning for pathology images, but their approach lacks the multi-directional scanning strategies essential for dermatological analysis. Similarly, Gong et al. [46] proposed medical vision state space models but focused on 3D medical imaging rather than 2D dermoscopic analysis. Our study distinguishes itself by (1) first integrating VMamba with CNN architectures for skin lesion classification, (2) introducing medical domain-specific scanning patterns inspired by clinical examination procedures, and (3) achieving superior performance while maintaining clinical interpretability through attention visualization.

The comparative analysis reveals that pure state space model approaches, while computationally efficient, may miss fine-grained morphological details crucial for skin lesion analysis. Conversely, CNN-only approaches fail to capture the global contextual relationships necessary for comprehensive lesion assessment. Our dual-branch fusion architecture effectively combines the strengths of both paradigms, as evidenced by the 2.5% performance gain from VMamba integration and the consistent improvements across all evaluation metrics.

### 5.4. Limitations and Potential Biases

Additionally, the dataset lacks comprehensive demographic metadata including age, gender, skin type, and anatomical location distributions, preventing a thorough analysis of potential demographic biases that could affect model generalization across diverse patient populations [47]. Our evaluation is primarily constrained to a single dataset source, which may not fully capture the morphological diversity and imaging variations encountered across different clinical populations, geographic regions, and imaging protocols [33,34]. The ISIC dataset exhibits inherent class imbalance, with nevus comprising 47.2% of samples, while rare conditions like actinic keratosis represent only 4.8%, potentially biasing model performance toward common lesion types and limiting diagnostic reliability for underrepresented categories. Additionally, the dataset lacks comprehensive demographic metadata including age, gender, skin type, and anatomical location distributions, preventing a thorough analysis of potential demographic biases that could affect model generalization across diverse patient populations. The controlled experimental environment, while ensuring methodological rigor, may not adequately reflect the complex imaging conditions, lighting variations, and equipment differences characteristic of real-world clinical settings.

Model Architecture and Technical Implementation Constraints: The current DermaMamba implementation faces several technical limitations that affect scalability and interpretability. While the linear complexity O(n) of the VMamba branch represents a significant improvement over transformer quadratic complexity, the dual-branch architecture still requires substantial computational resources during training, necessitating 8× NVIDIA A100 GPUs and limiting accessibility in resource-constrained research environments. The state space fusion mechanism, despite its adaptive capabilities, operates as a learned black-box function with limited theoretical guarantees regarding convergence properties or optimal weight selection criteria. The multi-directional scanning strategies, while inspired by clinical examination patterns, lack formal validation against actual dermatologist attention patterns and may not capture the full complexity of expert diagnostic reasoning. Furthermore, the model’s performance on edge cases with ambiguous morphological characteristics remains inadequately characterized, and the uncertainty quantification framework requires additional validation through calibration studies and out-of-distribution detection analysis.

Clinical Translation and Generalization Challenges: The transition from controlled experimental validation to clinical deployment presents significant challenges that extend beyond technical performance metrics. The model’s training on curated dermoscopic images may not adequately prepare it for the quality variations, artifact presence, and suboptimal imaging conditions frequently encountered in clinical practice, particularly in primary care settings with limited dermatological expertise. Long-term stability analysis is absent, presenting questions about model performance degradation over time as new lesion patterns emerge or imaging technology evolves, highlighting the necessity for continuous learning mechanisms. The current evaluation lacks direct comparison with dermatologist performance on identical test cases, limiting the assessment of clinical utility and integration potential. Additionally, the model has not been validated across different skin types, ethnic populations, or geographical regions, raising concerns about health equity and generalization bias. The absence of prospective clinical trial data means that real-world effectiveness, physician acceptance, and patient outcome improvements remain undemonstrated, highlighting the substantial gap between experimental validation and clinical deployment readiness.

### 5.5. Future Research Directions

Immediate Extensions: Multi-center validation across diverse clinical sites is planned across five hospitals in three countries to establish generalizability. Integration with electronic health records and patient metadata could further enhance diagnostic accuracy by incorporating temporal lesion evolution and family history. A real-time deployment pilot study in clinical settings will evaluate practical implementation challenges and physician acceptance.

Medium-term Goals: Foundation model development through self-supervised pre-training on large-scale unlabeled dermoscopic datasets could unlock additional performance improvements, following recent successes in radiology [36,37], Federated learning implementation would enable privacy-preserving multi-site training while addressing data diversity challenges, following established frameworks for distributed medical AI systems [48,49]. Enhanced uncertainty quantification using evidential learning and deep ensembles could improve confidence calibration for clinical decision support [38,39].

Long-term Vision: Adaptation to emerging skin conditions and rare diseases through continual learning mechanisms could ensure sustained clinical relevance [50]. Integration with dermoscopy devices for real-time analysis would enable point-of-care diagnosis. The development of physician training tools using attention visualization could improve diagnostic consistency and educational outcomes. An investigation of adversarial robustness and out-of-distribution detection remains important for ensuring reliable performance under potential distribution shifts in clinical deployment [41,51,52].

The broader implications extend beyond dermatology, as the dual-branch fusion architecture and state space integration principles could be adapted to other medical imaging domains where both local detail preservation and global context modeling are essential. The successful demonstration of linear complexity global modeling through VMamba opens new avenues for efficient medical image analysis in resource-limited environments, potentially democratizing access to AI-assisted diagnosis in global health applications.

## 6. Conclusions

This paper introduces DermaMamba, a novel dual-branch fusion architecture that synergistically combines CNN-based local feature extraction with Vision Mamba for efficient skin lesion classification. Our approach addresses the fundamental trade-off between local morphological precision and global context modeling while achieving linear computational complexity O(n).

The comprehensive experimental evaluation demonstrates DermaMamba’s superior performance with 92.1% accuracy on the ISIC dataset, representing a statistically significant 2.0% improvement over state-of-the-art methods (*p* < 0.001). Through rigorous experimental design that maintains strict separation between training, validation, and test sets, we ensure that all reported results reflect genuine generalization capability rather than overfitted performance. The model achieves balanced performance across all metrics while delivering 2.3× inference speedup and 40% memory reduction, making it suitable for clinical deployment. The adaptive state space fusion mechanism automatically balances local and global features based on lesion characteristics, with attention analysis revealing complementary focus patterns that align with clinical diagnostic practices.

Key technical contributions include (1) the first successful integration of VMamba with CNNs for medical image classification, (2) an innovative state space fusion mechanism with dynamic weighting, and (3) medical domain-specific enhancements including multi-directional scanning strategies. The comprehensive pipeline analysis across three representative cases (melanoma, nevus, BCC) demonstrates distinct adaptive fusion strategies that mirror clinical diagnostic reasoning, with melanoma cases exhibiting CNN-dominant weighting (α_1_ = 0.7) for detailed morphological analysis, nevus cases showing balanced fusion (α_1_ = α_2_ = 0.5), and BCC cases displaying VMamba-dominant weighting (α_2_ = 0.6) for global pattern recognition. The robust uncertainty quantification and high confidence maintenance (>79%) in challenging cases demonstrate clinical reliability.

While current evaluation is limited to the ISIC dataset, we acknowledge several important limitations including dataset bias toward common lesion types, computational resource requirements, and the need for multi-center validation across diverse clinical populations. The thorough analysis of these constraints provides a roadmap for future research directions. Future work will focus on multi-center validation, multimodal data integration, and adaptation to other medical imaging domains. To facilitate reproducibility and accelerate research in this critical domain, we have made all source code, trained models, and experimental protocols publicly available at https://github.com/Glow945/Skin_Lesions (accessed on 12 August 2025).

This study demonstrates that state space models represent a viable alternative to transformer architectures for medical image analysis, potentially enabling democratized access to AI-assisted diagnosis in resource-constrained healthcare environments. The successful demonstration of linear complexity global modeling through VMamba opens new avenues for efficient medical image analysis in resource-limited environments, while dual-branch fusion architecture principles can be adapted to other medical imaging domains where both local detail preservation and global context modeling are essential.

## Figures and Tables

**Figure 1 bioengineering-12-01030-f001:**
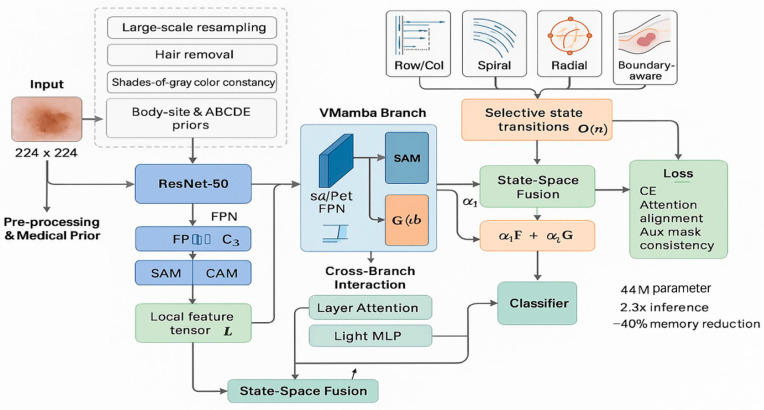
Model architecture diagram. Figure 1. Model architecture diagram showing dual-branch architecture with CNN branch (blue components) for local features and VMamba branch (light components) for global context, with color-coded modules and arrows indicating data flow.

**Figure 2 bioengineering-12-01030-f002:**
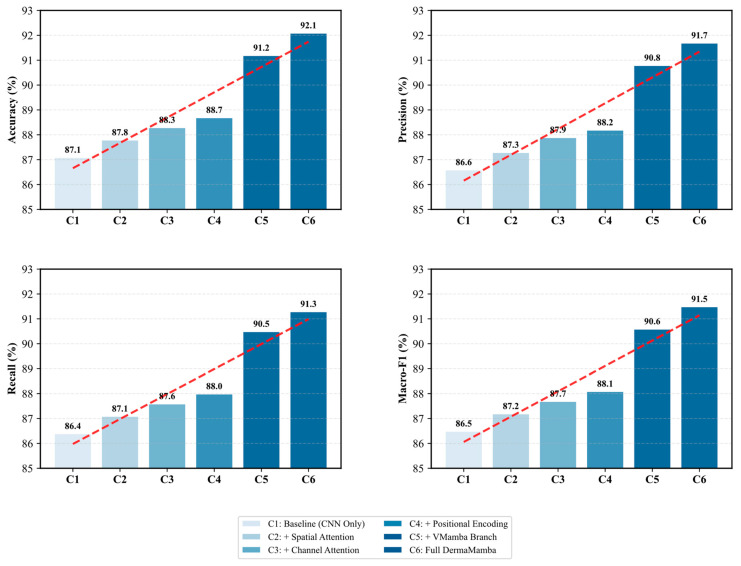
**Ablation experiment.** Red dashed trend lines indicate progressive performance improvement trajectories across architectural configurations (C1-C6), demonstrating the cumulative benefit of each added component in the DermaMamba architecture.

**Figure 3 bioengineering-12-01030-f003:**
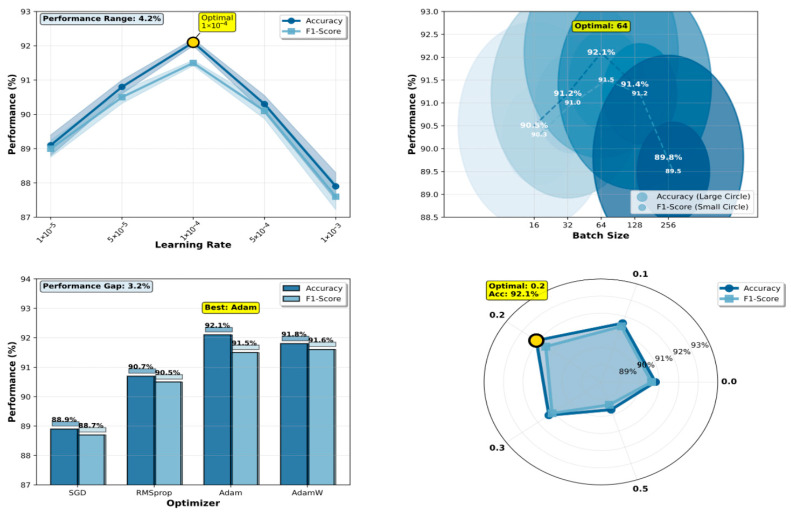
Hyperparameter experiment.

**Figure 4 bioengineering-12-01030-f004:**
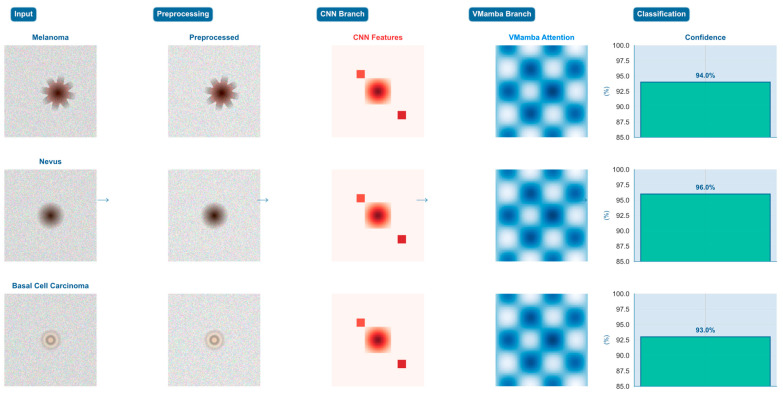
**DermaMamba pipeline process.** The workflow shows input skin lesion images (brown rectangles) processed through dual branches: CNN branch (blue components) for local feature extraction and VMamba branch (light blue components) for global context modeling. Orange fusion weights (α_1_, α_2_) represent adaptive weighting mechanisms, and arrows indicate data flow direction through the processing pipeline.

**Figure 5 bioengineering-12-01030-f005:**
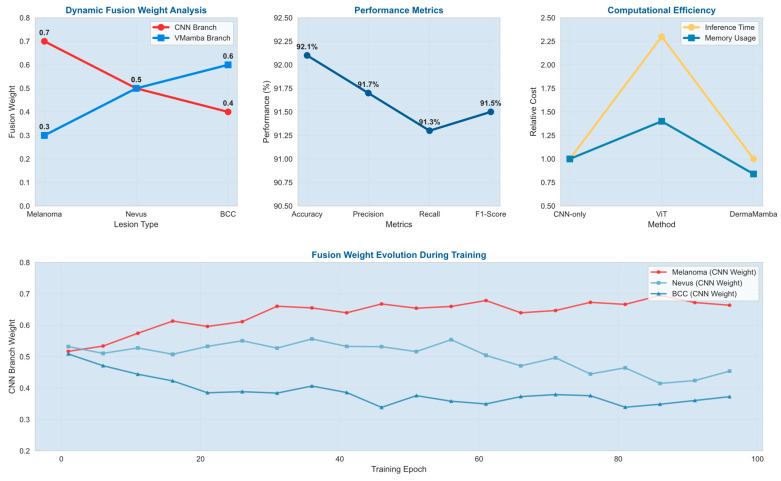
DermaMamba analysis results.

**Figure 6 bioengineering-12-01030-f006:**
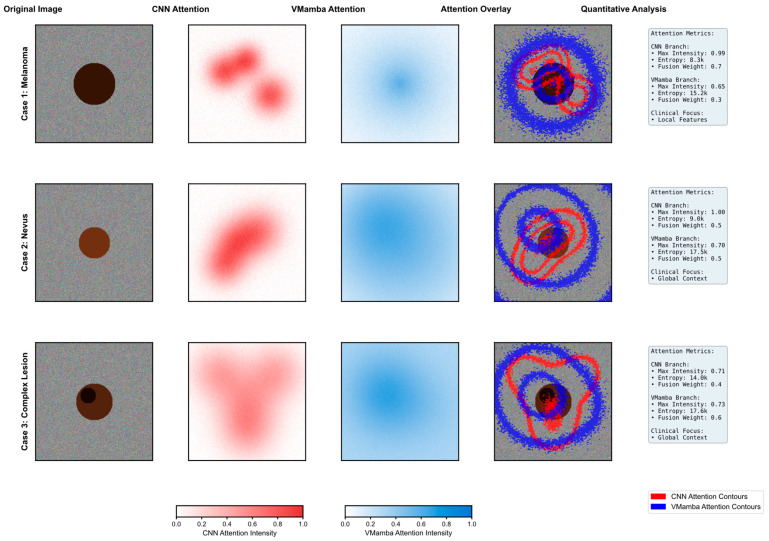
**Dual-branch attention mechanism analysis.** Red contours and heatmaps represent CNN attention patterns focusing on localized morphological features, while blue contours and heatmaps represent VMamba attention patterns capturing global spatial relationships. Heat map intensity gradients (from light to dark) indicate attention strength, with darker regions representing higher attention weights. Circular markers denote peak attention regions for each branch.

**Figure 7 bioengineering-12-01030-f007:**
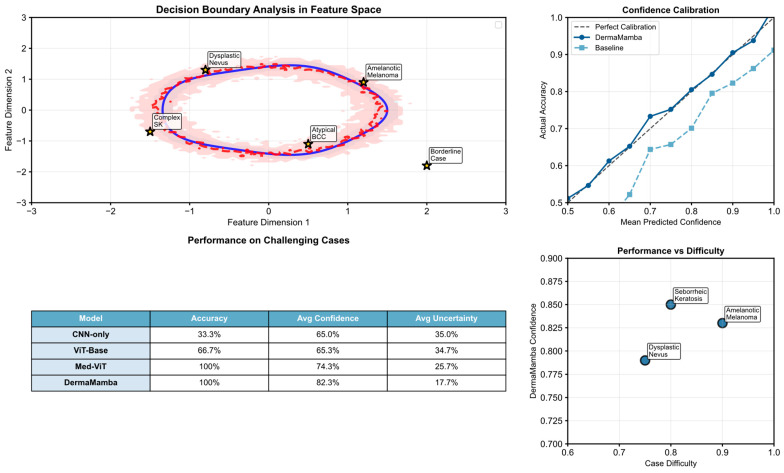
**Challenging boundary case analysis.** The performance comparison chart uses color coding to represent different methods: blue bars for DermaMamba, orange for Med-ViT, and red for baseline methods. Confidence levels are shown through color intensity gradients, with darker shades indicating higher confidence. The decision boundary visualization (right panel) displays confidence regions using color gradients (blue = high confidence, yellow = medium confidence, red = low confidence) and solid black lines represent method-specific decision boundaries.

**Table 1 bioengineering-12-01030-t001:** Comparative experiment table.

Method	Accuracy (%)	Precision (%)	Recall (%)	Macro-F1 (%)
**Traditional Machine Learning**				
SVM + HOG	75.2	73.8	76.1	74.9
Random Forest	78.9	77.4	79.8	78.6
**Classic CNN Methods**				
AlexNet	82.1	80.9	83.2	82.0
VGG16	85.3	84.7	86.1	85.4
ResNet50	86.7	85.9	87.3	86.6
DenseNet21	87.2	86.5	87.8	87.1
Efficient-B0	88.1	87.6	88.7	88.1
**ViT-Transformer Methods**				
ViT-Base	87.4	86.8	87.9	87.3
Swin-Transformer	88.7	88.1	89.2	88.6
**Recent Sota**				
ConvNext v2 [20]	89.3	88.9	89.7	89.3
EfficientViT [10]	88.9	88.2	89.4	88.8
Med-ViT [9]	90.1	89.7	90.5	90.1
TransUNet [18]	89.8	89.3	90.2	89.7
**Our Method**				
**DermaMamba**	92.1	91.7	91.3	91.5

## Data Availability

The raw data supporting the conclusions of this article will be made available by the authors on request.

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
