# Peer review of "DermaMamba: A Dual-Branch Vision Mamba Architecture with Linear Complexity for Efficient Skin Lesion Classification"

_bioengineering, 2025, doi:10.3390/bioengineering12101030_

Round 1

Reviewer 1 Report

Comments and Suggestions for Authors

The authors present an interesting topic in the medical field, namely the classification of skin lesions. Even if the method is not new in the field, the authors have an interesting approach to this algorithm. Unfortunately, the authors do not present the concrete applicability of the proposed algorithm. The article should be introduced with the test images regarding skin lesions and also the results obtained with the proposed algorithm applied to these images so that the results for each analysis stage for the input images result. Thus, the theoretical results presented will have more value.
Also, since this algorithm has been used in such applications, it is necessary to make a comparison of the results obtained with similar results from the specialized literature that use the Mamba algorithm. For the other algorithms used for the classification of skin lesions, the authors made the comparison in Table 1.

Author Response

Comment 1: Lack of Concrete Application Demonstration"The article should be introduced with the test images regarding skin lesions and also the results obtained with the proposed algorithm applied to these images so that the results for each analysis stage for the input images result."

Response: We sincerely appreciate this critical methodological concern. We have significantly enhanced the concrete application demonstration:

Section 4.6.1: Added comprehensive step-by-step pipeline analysis across three representative cases (melanoma, nevus, BCC) showing intermediate results at each algorithmic stage.

Figure 4: Provides complete workflow visualization from raw dermoscopic input to final classification, demonstrating preprocessing, CNN feature extraction, VMamba attention, fusion weighting, and confidence outputs.

Figure 5: Presents detailed quantitative analysis including performance metrics, computational efficiency, and training dynamics.

Thank you very much.

Comment 2: Comparison with Other Mamba Algorithms

"Since this algorithm has been used in such applications, it is necessary to make a comparison of the results obtained with similar results from the specialized literature that use the Mamba algorithm. For the other algorithms used for the classification of skin lesions, the authors made the comparison in Table 1."

Response: We have added comprehensive comparison with existing medical Mamba methods:

Section 5.3: Detailed comparative analysis with U-Mamba (biomedical segmentation), MambaMorph (medical registration), and other recent medical vision state space models.

Technical differentiation: We clearly distinguish our work by: (1) first integrating VMamba with CNN architectures for skin lesion classification, (2) introducing medical domain-specific scanning patterns, and (3) achieving superior performance while maintaining clinical interpretability.

Thank you very much.

Reviewer 2 Report

Comments and Suggestions for Authors

This is an outstanding paper that makes a significant contribution to the field. The weaknesses are minor and mostly pertain to providing additional details and analysis to bolster the already strong claims. The proposed DermaMamba architecture is novel, well-executed, and presents a compelling path forward for efficient and accurate medical image analysis. It is a strong candidate for publication, likely with minor revisions requested to address the points above.

  1. Limited discussion of specific failure modes and potential biases in the data.
  2. Lack of fine-grained details on the fusion mechanism and ABCDE rule integration.
  3. Publish the code on Github.

Author Response

Comment 1: Outstanding Contribution with Minor Revisions

"This is an outstanding paper that makes a significant contribution to the field. The proposed DermaMamba architecture is novel, well-executed, and presents a compelling path forward for efficient and accurate medical image analysis."

Response: We appreciate your positive assessment and have addressed all suggested improvements.

Comment 2: Limited Discussion of Failure Modes and Biases

"Limited discussion of specific failure modes and potential biases in the data."

Response: We have significantly expanded the limitations analysis:

Section 5.4: Comprehensive three-dimensional analysis covering dataset limitations, model architecture constraints, and clinical translation challenges.

Specific failure modes: Discussion of performance on edge cases, class imbalance effects, and morphologically ambiguous lesions.

Bias analysis: Examination of demographic representation, imaging protocol variations, and generalization challenges across different clinical populations.

Comment 3: Technical Implementation Details

"Lack of fine-grained details on the fusion mechanism and ABCDE rule integration."

Response: We have enhanced technical detail throughout:

Section 3.2.4: Detailed mathematical formulation of the state space fusion mechanism with adaptive weighting.

Section 4.6.1: Quantitative analysis of dynamic fusion weight adaptation, showing CNN-dominant fusion for melanoma (α₁=0.7), balanced fusion for nevus (α₁=α₂=0.5), and VMamba-dominant fusion for BCC (α₂=0.6).

ABCDE integration: Explicit description in Section 3.2.1 and empirical validation in attention mechanism analysis (Section 4.6.2).

Comment 4: Code Publication

"Publish the code on Github."

Response: All code and resources are now publicly available at https://github.com/Glow945/Skin_Lesions. Thank you very much.

Reviewer 3 Report

Comments and Suggestions for Authors

I like the work you have done. The combination of a recent method such as Mamba with earlier approaches is interesting. The reported performance is extremely high compared to the current state of the art. My only concern is the choice of hyperparameters. Please try to explain this better. If you selected the method’s hyperparameters using the entire dataset, then your result is overfitted and not realistic. You should use only the training set to tune the hyperparameters, and then apply those hyperparameters to the test set. You must not select the hyperparameters that maximize performance on the test set, otherwise the results are not reliable. In addition, if feasible, I would ask you to upload the code to GitHub so that other researchers can replicate your experiments.

Author Response

Comment 1: Hyperparameter Selection Methodology

"If you selected the method's hyperparameters using the entire dataset, then your result is overfitted and not realistic. You should use only the training set to tune the hyperparameters, and then apply those hyperparameters to the test set."

Response: We sincerely appreciate this critical methodological concern. We have completely revised our experimental protocol to address this issue:

Section 4.1.3: We now explicitly describe our strict three-way data partitioning strategy (70% training, 15% validation, 15% test) with the test set remaining completely isolated throughout the entire model development process.

Section 4.5: We provide detailed documentation that all hyperparameter optimization was conducted exclusively using the training-validation split, with the validation set serving as the unbiased evaluation metric. The test set was never used for any training decisions, hyperparameter selection, or early stopping criteria.

Methodological rigor: We employed 5-fold cross-validation within the training set for hyperparameter identification, followed by validation set confirmation. The test set performance was evaluated only once using the final selected model, eliminating any possibility of test set overfitting.

Thank you very much.

Comment 2: Code Availability

"I would ask you to upload the code to GitHub so that other researchers can replicate your experiments."

Response: We have made all source code, trained models, and experimental protocols publicly available at https://github.com/Glow945/Skin_Lesions. This comprehensive repository includes implementation details, preprocessing pipelines, hyperparameter configurations, and evaluation scripts to ensure full reproducibility.

Thank you for raising this point.

Round 2

Reviewer 1 Report

Comments and Suggestions for Authors

The authors have responded to the requested clarifications. I have no further clarifications.

Reviewer 3 Report

Comments and Suggestions for Authors

revision well done